# Brain-to-Text Decoding with Context-Aware Neural Representations and Large Language Models

## Abstract

Decoding attempted speech from neural activity offers a promising avenue for restoring communication abilities in individuals with speech impairments. Previous studies have focused on mapping neural activity to text using phonemes as the intermediate target. While successful, decoding neural activity directly to phonemes ignores the context dependent nature of the neural activity-to-phoneme mapping in the brain, leading to suboptimal decoding performance. In this work, we propose the use of diphone - an acoustic representation that captures the transitions between two phonemes - as the context-aware modeling target. We integrate diphones into existing phoneme decoding frameworks through a novel divide-and-conquer strategy in which we model the phoneme distribution by marginalizing over the diphone distribution. Our approach effectively leverages the enhanced context-aware representation of diphones while preserving the manageable class size of phonemes, a key factor in simplifying the subsequent phoneme-to-text conversion task. We demonstrate the effectiveness of our approach on the Brain-to-Text 2024 benchmark, where it achieves state-of-the-art Phoneme Error Rate (PER) of 15.34% compared to 16.62% PER of monophone-based decoding. When coupled with finetuned Large Language Models (LLMs), our method yields a Word Error Rate (WER) of 5.77%, significantly outperforming the 8.93% WER of the leading method in the benchmark.

## 1 Introduction

Verbal communication is a unique feature of human social interaction. Loss of ability to articulate speech as a result of neurological pathologies such as stroke and Amyotrophic Lateral Sclerosis (ALS) can significantly reduce the quality of life for affected individuals. Recent advancements in Brain-Computer Interfaces (BCI) offer promising pathways toward restoring communication ability in these patients by translating neural activity into communicative messages. These messages can be conveyed through various modalities, including typed characters (Pandarinath et al., 2017), handwriting (Willett et al., 2021), text (Herff et al., 2015; Willett et al., 2023a; Metzger et al., 2023), and synthesized speech (Metzger et al., 2023).

Among existing speech BCI systems, the methods with highest decoding accuracy and throughput are those that translate neural signals associated with orofacial movements during attempted speech into fundamental acoustic units (phonemes), which are then decoded into words and sentences (Willett et al., 2023a; Metzger et al., 2023). This two-staged approach typically involves (1) neural signal to phonemes: using a temporal deep network to decode a binned multi-channel neural time series into probability of phonemes being spoken at each time step, and (2) phonemes to text: employing a language model (LM) to infer the most probable sequence of words given the phoneme probabilities.

Prior work shows that decoding phonemes as an intermediate representation rather than directly decoding words, provides the system the flexibility to decode phrases from extensive vocabularies a limited set of training examples (Metzger et al., 2023), since from a fixed set of 40 phonemes, one can practically construct any word of any arbitrary length. This scalability is especially advantageous given the limited availability of neural recordings in clinical settings.

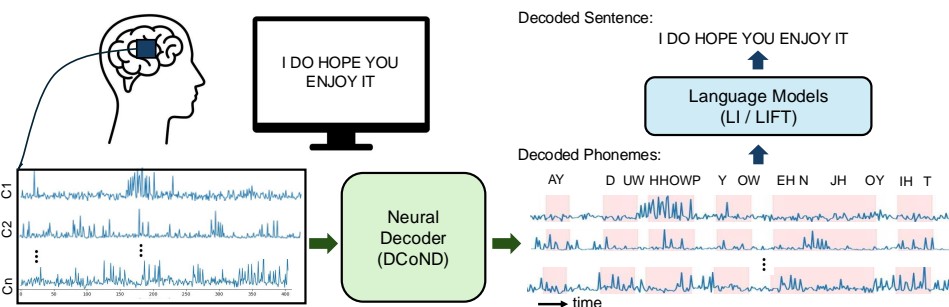

Figure 1: Overview of the Brain-to-Text decoding pipeline. The Neural Decoder with Divide-and-Conquer Strategy (DCoND) decodes multi-channel neural activity into phonemes. The phonemes are subsequently converted into words by LLMs using either ICL or fine-tuning techniques.

While decoding single phonemes from neural activity may offer more scalability than decoding words, it remains a challenging task. Given the innate variability of neural signals, the mapping from neural activity to phonemes is many-to-one and highly nonlinear. Furthermore, evidence suggests that cortical activation patterns producing a particular phoneme is not static, but can vary depending on the context of surrounding phonemes, a phenomenon known as *coarticulation* (Bouchard & Chang, 2014; Mugler et al., 2014). In other words, cortical neurons at any given time during speech production are likely encoding a phoneme along with its context, rather than a phoneme in isolation. Given this observation, diphone (Nedel et al., 2000) - a sequence of two adjacent phonemes - is a more suitable representation for capturing this context dependency in neural signals and potentially reducing the nonlinearity in phoneme decoding. Hence we propose to decompose the phoneme classification task into subtasks of diphone classification, after which diphone probabilities are summed up to obtain the phoneme prediction, i.e. predicting phoneme distribution by marginalizing over the diphone distribution. We show that this divide-and-conquer strategy significantly enhances phoneme decoding performance.

Recently introduced approaches leverage language models, such as n-gram model, to translate phoneme probabilities into words (Willett et al., 2023a; Metzger et al., 2023; Benster et al., 2024). Notably, (Benster et al., 2024) further uses GPT3.5 (Brown et al., 2020) after a 5-gram model to refine the resulting word sequences into coherent sentences by ensembling multiple 5-gram transcription candidates. However, the transcription candidates generated by the n-gram model can significantly deviate from the ground truth phoneme sequence. To address this issue, we propose to augment the ensembling method in (Benster et al., 2024) to include decoded phonemes alongside transcription candidates, which proves to provide extra information for GPT3.5 to infer the correct transcription. Additionally, we propose an In-Context Learning (ICL) paradigm for LLMs, enabling them to adapt quickly to newly decoded inputs in a gradient-free manner without the need for the computationally expensive finetuning process. This approach offers a more efficient alternative for improving transcription accuracy in resource-constrained settings.

In summary, our contributions in this work are as follows:

- We propose DCoND (**D**ivide-and-**Co**nquer **N**eural **D**ecoder), a novel framework for decoding phonemes from neural activity during attempted speech. Backed by neuroscientific insights, DCoND infers the temporal phoneme distribution by marginalizing over the diphone distribution, leveraging the context-dependent nature of phonemes in neural representation.

- We propose incorporating decoded phonemes alongside decoded words in an **LLM**-based ensembling strategy to enhance the speech decoding performance. We also propose the use of (**ICL**) paradigm (DCoND-LI) as an alternative to **F**ine**T**uning LLMs (DCoND-LIFT), offering a more efficient solution for resource-constrained brain-to-text systems.

- We demonstrate the effectiveness of our approaches on the Brain-to-Text 2024 benchmark, where our approach achieves state-of-the-art (SOTA) PER of 15.34% and WER of 5.77%, a significant improvement compared to 8.93% WER of the leading SOTA method.

## 2 RELATED WORK

**Brain-to-Text Decoding with Speech Waveforms**   The problem of decoding speech from neural activity is relatively more manageable when the temporal correspondence between the neural signal and the speech is known. Such a situation occurs during speech perception tasks (Poeppel et al., 2008; Défossez et al., 2023; Fodor et al., 2024; Yang et al., 2024). In this case, the mapping from neural activity to perceived speech could be learned through supervised learning (Fodor et al., 2024; Yang et al., 2024) or contrastive learning (Défossez et al., 2023). Temporal correspondence between neural activity and speech also exists in speech production experiments performed by individuals who still retain the ability to speech normally, during which concurrent speech waveforms are recorded. Studies for such scenarios include (Jou et al., 2006; Schultz & Wand, 2010; Kapur et al., 2018; Meltzner et al., 2018; Diener et al., 2018; Janke & Diener, 2017; Chen et al., 2024). When the produced speech is not fully observed, Gaddy and Klein propose to use dynamic time warping and canonical correlation analysis to align the neural signals with recorded audio signal(Gaddy & Klein, 2020; 2021). In contrary to these works, our study focuses on speech decoding when audio recordings of speech are not available.

**Brain-to-Text Decoding without Speech Waveforms**   In cases of individuals who cannot produce intelligible speech, the speech decoding problem could be entirely avoided by using typing-based systems, albeit with low throughput (Vansteensel et al., 2016; Pandarinath et al., 2017; Linse et al., 2018). Early works on speech decoding were demonstrated with a small vocabulary size (Moses et al., 2021; Kellis et al., 2010), which could be improved by learning to decode letters(Metzger et al., 2022). Other studies investigate phonemes as the decoding target (Pei et al., 2011; Mugler et al., 2014; Herff et al., 2015; Willett et al., 2023a; Metzger et al., 2023). However, decoding phonemes directly can be a difficult task since neural representations for phonemes could change depending on the contexts they are spoken(Mugler et al., 2014). We leverage this observation to devise our strategy using diphones as decoding target.

**Brain-to-Text Decoding vs. Speech-to-Text Decoding**   While there are similarities between brain-to-text and speech-to-text decoding, decoding text from neural signals is a significantly more challenging task. One key difference is that speech signals are univariate, while neural activity is multivariate as it is recorded by multi-channel electrodes. Furthermore, neural signal is far more intricate. Less is known about how neurons encode speech within their spiking activity, as well as the degree to which speech-relevant components can be extracted from the complex interaction of neural population. However, brain-to-text decoding methods have drawn inspiration from speech-to-text decoding research, commonly referred to as Automatic Speech Recognition (ASR). Earlier studies (Miao et al., 2015; Aggarwal & Dave, 2011; Huang et al., 2014) use Hidden Markov Models and Gaussian Mixture Models to decode recorded speech signals into phonemes before translating into words. (Darjaa et al., 2011; LAleye et al., 2016) suggest that using diphone or triphone could enhance the accuracy of ASR systems. Modern ASR systems have transitioned to end-to-end learning approaches, directly decoding speech signals into words (Prabhavalkar et al., 2023; Graves, 2012; Gulati et al., 2020; Hsu et al., 2021; Schneider et al., 2019). However, end-to-end learning requires a large number of word targets which are generally not available in neuroscience domain. We therefore adopt the two-stage system for brain-to-text decoding, where phonemes serve as the intermediate decoding targets.

**In-Context Learning**   LLMs pretrained on large corpora of texts exhibit the ability to learn new tasks in-context (Brown et al., 2020). That is, conditioning on a few demonstrations of input-target pairs, LLMs can generalize to unseen cases without updating their weights. This ICL ability has proven useful across a wide range of tasks (Wei et al., 2022; Touvron et al., 2023). While ICL typically underperforms a specialized LLM finetuned for a specific downstream task, it still surpasses zero-shot inference, and is particularly valuable when finetuning is not feasible due to resource constraints such as time or computational power, or the inacessibility of proprietary LLMs (Mosbach et al., 2023).

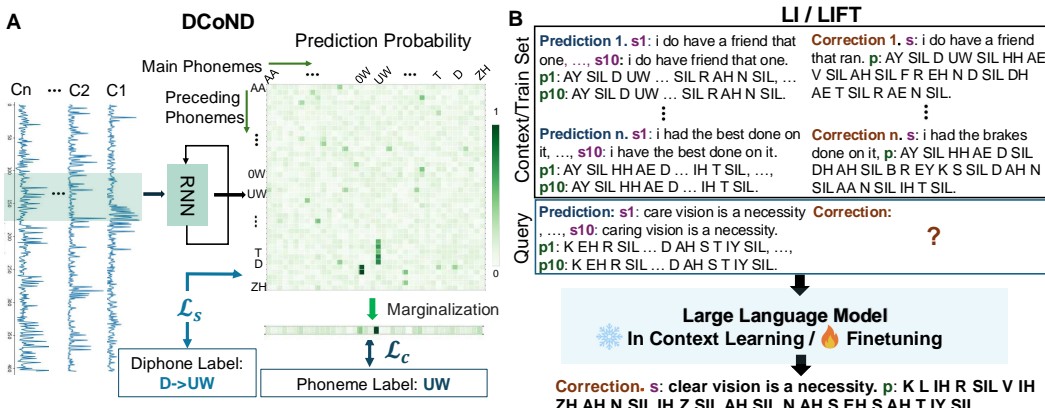

Figure 2: **A**: Illustration of the brain-to-phoneme decoding pipeline (DCoND). An RNN in DCoND takes multi-channel neural signals as inputs and generates diphone probabilities, which are then marginalized into single phoneme probabilities. **B**: Illustration of the ensembling method for refining transcription predictions (LI/LIFT). Given an ensemble of phoneme and transcription candidates as a query, GPT3.5 produces the most sensible transcription composed from these inputs. To do this, the LLM leverages examples of prediction-correction pairs provided either in-context at inference time (LI) or as training data during the finetuning process (LIFT).

## 3 METHODS

**Problem formulation**   The problem of decoding phonemes from neural activity can be formulated as follows. Let $f : X \to Z$ be the mapping from neural activity $X \in \mathbb{R}^{T \times D}$ to phoneme sequence $Z \in \mathbb{Z}^{T'}$, where $D$ is the number of neural features, $T$ is the number of neural time bins, and $T'$ is the number of ground truth phonemes in a sentence. We note that $T > T'$ in general, i.e. the articulation of one phoneme may span multiple timesteps. We also emphasize that there is no ground truth temporal alignment between $X$ and $Z$ due to the nature of the silent speech task. Both $T$ and $T'$ vary across trials depending on the length of the sentence in that trial. We aim to learn a model $f_\theta : X \to Z$ to approximate $f$ with a set of parameters $\theta$. We use an RNN model (GRU) for $f_\theta$ **together with Connectionist Temporal Classification (CTC) loss as the optimization objective**. GRU has demonstrated superior performance on this dataset, as reported in previous works (Willett et al., 2023a; Benchetrit et al., 2023). A comparative study of alternative architectures, such as LSTM and transformer, is avaliable in the Appendix. Decoded phonemes $Z$ can be subsequently translated to sentences $Y$ with the help of a language model $h_\phi : Z \to Y$, where $h_\phi$ can be a pre-built statistical language model, e.g. 5-gram, or an LLM, e.g. GPT3 (Brown et al., 2020). The overall pipeline is depicted in Figure 1.

**A Divide-and-Conquer strategy for phoneme decoding**   Decoding phonemes from neural activity is a nontrivial task given the highly nonlinear nature of $f$ and the variability of the neural population dynamics. Evidence exists that the neural representations for phonemes vary depending on the surrounding contexts (Bouchard & Chang, 2014; Mugler et al., 2014). We illustrate this observation in Figure 3 where segments of phoneme-aligned neural activity form clusters in the neural space based on the context they are in. It can be seen that there is no single cluster representing each phoneme, but rather each phoneme is represented by multiple subclusters. We further show that the subclusters are identifiable by the phoneme preceding the phoneme of interest. For instance, the phoneme AH is represented by subclusters DH $\to$ AH and SIL $\to$ AH (see further discussion in Section 4.4). Learning to model these context-aware sub-units of speech instead of single phonemes directly could facilitate the phoneme decoding task. Concretely,

$$f(x) := p(Z|X) = \sum_S p(Z, S|X) = \sum_{s \in S} g_s^Z(x) \tag{1}$$

where $S$ is a random variable denoting the context surrounding the phoneme $Z$. **For simplicity of notation, here we consider the prediction of $Z$ at single time step, i.e. $T' = 1$. $Z$ takes discrete**

**values from phoneme classes, i.e.** $Z \in [1, C]$**.** The problem of learning single phoneme classes ($f$) now reduces to the problem of learning the phoneme context-dependent subclasses ($g_s^Z$), which is more manageable and in-line with the context-dependent nature of the data. We refer to our phoneme decoder with this divide-and-conquer strategy as **DCoND**.

**Diphone as a context-dependent representation of phonemes**  The context-dependent subclasses could be defined in multiple ways. In this work, we adopt diphone, a context-dependent representation for phoneme sequences where transitions between phonemes are the subject of interest. For example, the single phoneme representation of "hope", $H, \quad OW, \quad P$, will have a diphone representation:

$$SIL \to H, \quad H \to H, \quad H \to OW, \quad OW \to OW, \quad OW \to P, \quad P \to P, \quad P \to SIL.$$

where 'SIL' indicates the silence between the words. Diphone expands the length of phoneme sequence to $T'' = 2T'$ and increases the number of decoding classes to $C^2$, where $C = 40$ for the English language[1].

Formally, we reformulate the problem of decoding phoneme from neural activity as the marginalization over the distribution of diphones, conditioning on the observed neural activity

$$p(Z = c_i | X) = \sum_{c_j \in S} p(c_j, c_i | X),$$

where $p(c_j, c_i | X)$ is the probability of neural activity X encoding the diphone $c_j \to c_i$. A visualization of the marginalization process is shown in Fig. 2A. Neural activity is processed by an RNN to predict the probability of $40^2$ diphones being spoken at each timestep. The diphone probability is depicted by a $40 \times 40$ matrix where columns correspond to the main phonemes and rows correspond to the preceding phonemes. The single phoneme probability is then obtained by summing the joint probabilities column-wise.

**Parameter Optimization for Phoneme Decoding**  As mentioned above, we do not have the temporal alignment between $T$ timesteps of neural activity and $T'$ ground truth phonemes in each trial. We therefore use the Connectionist Temporal Classification (CTC) loss as proposed in (Graves et al., 2006) to resolve the non-alignment issue. Specifically, we try to maximize the probability of Z given X

$$p(Z|X) = \sum_{A \in \mathcal{A}_{(X,Z)}} \prod_{t=1}^{T'} p(a_t|X), \tag{2}$$

where $A_{(X,Z)}$ is the set of valid alignments between $X$ and $Z$.

Now that we have the diphone representation for each ground truth sentence, we consider the CTC losses over both the diphone and single phoneme representations:

$$\mathcal{L} = \alpha \mathcal{L}_c + (1 - \alpha)\mathcal{L}_s \tag{3}$$

where $\mathcal{L}_c = -\log(\sum_{A \in \mathcal{A}_{(X,Z)}} \prod_{t=1}^{T'} p_m(a_t|X))$ is the loss for single phoneme decoding, $\mathcal{L}_s = -\log(\sum_{A \in \mathcal{A}_{(X,S)}} \prod_{t=1}^{T''} p(a_t|X))$ defines the loss over subclasses (diphone) decoding.

Coefficient $\alpha$ controls the balance of the single phoneme decoding and diphone decoding. $\alpha$ is designed to be small at the beginning and gradually increase over the course of training. See Appendix A.6 for more implementation details.

**Word Decoding with Language Models**  The predicted phoneme probabilities are further transformed into high-quality text through (i) generation of transcription candidates from phonemes, (ii) re-scoring of transcription candidates, and (iii) error correction using an ensemble of selected candidates.

*Transcription Generation.*  During the phase of candidate sentence generation, we convert the predicted phoneme probabilities into words using a 5-gram model. Based on the predicted phoneme

---

[1]the phonemes are defined as per CMU Pronouncing Dictionary: http://www.speech.cs.cmu.edu/cgi-bin/cmudict/

probability distribution, the 5-gram model leverages its internal word and sentence distributions to generate the most likely sentence candidates (Miao et al., 2015; Willett et al., 2023a). Each candidate is associated with a likelihood score provided by the 5-gram model.

*Transcription Re-scoring* LLMs trained on large corpora of texts, such as the Open Pre-trained Transformer (OPT) (Zhang et al., 2022), could provide more accurate likelihood of the generated transcriptions. Hence, we use OPT to re-score the 5-gram likelihood outputs. The transcription candidates with the highest likelihoods are selected(Willett et al., 2023a).

*Transcription Error Correction with Ensemble Method* While the 5-gram and OPT models can correct some phoneme errors made by the phoneme decoder to produce more contextually sound sentences (transcriptions), these sentences are not always perfect. Variations of the phoneme decoding model could result in changes of generated and selected sentence candidates. Ensembles of phoneme decoding models, with each model being an expert in different situations, could mitigate the errors made by another model.

In (Benster et al., 2024) GPT3.5 is finetuned to evaluate an ensemble of 10 transcription candidates and generate the most sensible sentence from the 10 candidates. However, providing GPT3.5 only the candidate transcriptions hinders the LLM's ability to understand the underlying phoneme sequences, which are the generating source of the transcriptions and might have been incorrectly converted by the 5-gram model. We therefore propose to include both the transcription candidates and the corresponding phoneme sequences as inputs to GPT3.5, tasking the model with generating both the correct transcription and phoneme sequence. An illustration of such task is shown in Fig.2. By finetuning the LLM in this manner, we train it to infer the relationship between predicted phonemes and the predicted transcriptions, as well as identifying common model-specific mistakes made by the phoneme decoders across their predictions. We show in Section 4.3 that this strategy further boosts the WER from 8.06% to 5.77%.

In addition, since finetuning LLM is a resource-intensive process, we also propose to leverage ICL as an alternative learning paradigm for refining predicted transcriptions. Instead of finetuning GPT3.5 over multiple batches of ($10\times$ predictions, $1\times$ ground truth) pairs, we directly include $N$ examples of these pairs as context in each prompt, along with a query input to be refined. The LLM then leverages its ICL ability to quickly refine the query transcriptions without updating its weights. The prompts used for both in-context inference and finetuning are detailed in the Appendix A.8.

## 4 EXPERIMENTS

### 4.1 DATASET

We demonstrate the effectiveness of DCoND-LIFT in decoding attempted speech using the Brain-to-Text Benchmark 2024 (Willett et al., 2023a;b). The dataset was collected from a human subject with ALS who had lost the ability to produce intelligible speech. In the experiments, the subject attempts to silently speak sentences displayed on a screen. These sentences are composed from a vocabulary set of 125,000 words. In each trial, one sentence is shown followed by an auditory 'Go' cue, after which the subject attempts to speak at their own pace. Neural activity (multiunit threshold crossings and spike band power) is recorded from the ventral premotor cortex (6V) while the subject attempted speaking. Due to the nature of the silent speech task, the correspondence between neural activity and the produced speech is unknown. The dataset is split into training, validation, and competition sets with 8800, 600, and 1200 sentences, respectively.

### 4.2 EVALUATION METRICS

**PER** Phoneme Error Rate (PER) is calculated by comparing the decoded phoneme sequence with the ground truth phoneme sequence. After aligning the recognized phoneme sequence with the reference phoneme sequence, the number of insertions, deletions, and substitutions required to match the sequences are counted. The sum of these operations is divided by the total number of phonemes in the ground truth sequence to compute the PER. This metric reflects how accurately neural signals can be recognized into phonetic units.

Table 1: Performance comparison on Brain-to-Text 2024 Benchmark

| | PER×100 ↓ | WER×100 ↓ | P-WER×100 ↓ |
|---|---|---|---|
| NPTL (Willett et al., 2023a) | 16.62 | 9.46 | 11.33 |
| LISA (Benster et al., 2024) | – | 8.93 | – |
| DCoND-L (Ours) | **15.34** | 8.06 | **8.02** |
| DCoND-LI (Ours) | – | 7.29 | – |
| DCoND-LIFT (Ours) | – | **5.77** | – |

**WER** Similar to PER, word error rate (WER) is computed by aligning the sequence of recognized words with the ground truth sentence first and then counting the number of insertions, deletions, and substitutions of words needed to reconcile any discrepancies between the two sequences. The total number of these operations is divided by the total number of words in the reference sequence to obtain WER. As neural activity is translated into phonemes before converted into words, WER reflects the performance of both neural decoder and the language model.

**P-WER** We adapt Perceptual Word Error Rate (P-WER) (Metzger et al., 2023) to measure the quality of phoneme decoding at the word perception level. Specifically, we use eSpeak-NG (Reece H. Dunn)[2] to synthesize speech from the decoded phoneme sequences. Then the synthesized speech is translated into sentences by Whisper (Radford et al., 2022) from which the WER is estimated. Considering the systematic errors introduced by the eSpeak-NG synthesizer and the Whisper ASR system, we define P-WER as follows

$$\text{P-WER} = (1 - \frac{1 - \text{WER}_{Whisper\text{-}P}}{1 - \text{WER}_{Whisper\text{-}GT}}),$$

where $\text{WER}_{Whisper-GT}$ and $\text{WER}_{Whisper-P}$ are the WER measured on Whisper's decoded transcriptions when audio is synthesized with ground truth phoneme sequences (GT) and predicted phoneme sequences (P), respectively.

### 4.3 COMPARISON WITH SOTA METHODS

We show DCoND-LIFT achieves state-of-the-art performance on the Brain-to-Text Benchmark 2024, where WER is the primary evaluation metric (see Table 1). Specifically, we compared DCoND-LIFT with the leading methods NPTL (Willett et al., 2023a) and LISA (Benster et al., 2024). NPTL uses a 5-layer RNN to decode neural activity to phonemes, followed by a combination of 5-gram and OPT language models (Miao et al., 2015; Zhang et al., 2022) to translate decoded phonemes to texts. LISA uses the same RNN model architecture as NPTL to decode phonemes from neural activity, but leverages GPT3.5 to further improve transcriptions given by the 5-gram model. **See Appendix A.6 for more implementation details.**

As seen in Table 1, our model variants outperform the competing methods across the board. DCoND combined with 5-gram LM and OPT (DCoND-L) yields WER of 8.06%, compared to 9.46% WER of NPTL and 8.93% of LISA. Further sensitivity analysis is provided in Table 4 of the Appendix. Given that DCoND-L uses the same RNN backbone and LMs as NPTL, we posit that the improvements in WER come from the effectiveness of our divide-and-conquer phoneme decoding strategy. Indeed, DCoND-L achieves a better PER and P-WER (15.34% and 8.02% compared to 16.62% and 11.33% of NPTL), proving that modeling context-dependent phoneme representations facilitates the phoneme decoding task.

The WER further improves when we equip DCoND-L with the more powerful language model GPT3.5 to evaluate an ensemble of predicted transcriptions and their associated phoneme representations. When ensemble exemplars are shown to GPT3.5 in-context (DCoND-LI), WER improves from 8.06% to 7.29%. This performance is achieved with 25 ICL exemplars, the largest number of ICL exemplars GPT3.5 can afford due to its prompt length constraint. When we finetune GPT3.5 using all available training exemplars (DCoND-LIFT), WER is further boosted to 5.77%, a significant improvement over 8.93% WER of LISA. These results support our proposal of including both

---

[2]https://github.com/espeak-ng/espeak-ng

Table 2: Trade-offs between diphone loss and monophone loss.

| | $\alpha = 0.2$ | $\alpha = 0.4$ | $\alpha = 0.6$ (DCoND-L) | $\alpha = 0.8$ | $\alpha = 1.0$ (NPTL) |
|---|---|---|---|---|---|
| PER$\times 100 \downarrow$ | 15.64 | 15.26 | 15.34 | 15.49 | 16.62 |
| WER$\times 100 \downarrow$ | 8.47 | 8.70 | 8.06 | 8.64 | 9.46 |

transcriptions and phoneme representations in the demonstrations to GPT3.5 so that it can leverage the relationship between phonemes and words to refine the transcriptions.

### 4.4 PHONEME DECODING ANALYSES

**Neural activity represents phonemes in context-dependent clusters**  Previous works demonstrate that the accuracy of decoding phonemes from neural activity could degrade when phonemes are pronounced in the context of other phonemes as opposed to being pronounced individually (Mugler et al., 2014). To get a glimpse of how the brain encodes phonemes, in Fig. 3A we visualize phoneme-aligned segments of neural activity in the 2D t-SNE space (van der Maaten & Hinton, 2008). Since the dataset does not have the exact temporal correspondence between neural activity and phonemes, we leverage Dynamic Time Warping (DTW) to align the ground truth phonemes to neural activity segments according to the timestamps obtained from the decoded phonemes (Müller, 2007). We annotate the neural activity segments based on the resulting phoneme alignment. The visualization reveals that neural activity segments form distinct clusters in the t-SNE space. Notably, these clusters are organized based not only on single phonemes but also on the context in which they are spoken. For instance, during periods where 'T' is the main phoneme being spoken, the neural activity is organized into subclusters of AE→T (orange) and SIL→T (pink), depending on whether phoneme 'AE' or 'SIL' is spoken before 'T'. Similar observations hold for subclusters DH→AH (green) and SIL→AH (red) for phoneme 'AH'. We note that further subclusters could exist within each subcluster, suggesting a continuum of finer contexts beyond the preceding phoneme.

**Decoding diphone leads to enhanced clusters in latent space**  We visualize in Figures 3C and 3D the latent space at the last layer of the neural decoder when trained to decode single phonemes (monophones) vs. diphones. In Figure 3C, each color represents a single decoded phoneme label. For clear visualization, we selected five single phoneme classes with the most samples. The clusters that correspond to single phonemes appear to spread out over the whole space, and overlap with each other. In Figure 3D, each color represents a decoded diphone. Since there are fewer samples for each diphone, we visualize 16 diphone classes with the highest occurrence. It can be observed that the neural decoder represents diphones in the latent space by clusters that are significantly more condensed and well-separated. Such clear structure facilitates the subsequent classification of single phonemes and demonstrates the effectiveness of our divide-and-conquer phoneme decoding method.

**Phoneme Prediction Error Analysis**  In Figure 3B, we show the confusion matrix of the predicted phonemes and the ground truth phonemes. From the figure we can see that most phonemes are correctly classified with accuracy greater than 80%. The mistakes the model typically makes, if any, are on phonemes that are pronounced similarly. For example, the model usually confuses 'SH' with 'S', and 'CH' with 'TH'. Since the articulation of these phonemes is very similar, the neural activity generating them is likely to be similar. Such confusion is expected to some extent, given the ALS condition hindering the subject's ability to clearly articulate the desired words.

### 4.5 ABLATION STUDY

**Trade-off Between Diphone Loss and Monophone Loss**  We systematically investigate the trade-off between diphone loss $\mathcal{L}_c$ and monophone loss $\mathcal{L}_s$, controlled by the parameter $\alpha$ in Equation 3. The impact of varying $\alpha$ on model performance is shown in Table 2. We find that a balance between these two losses, with $\alpha = 0.6$, yields the most optimal results. Consequently, we adopt $\alpha = 0.6$ for all DCoND models used in this paper.

**Alternatives for context-dependent phoneme representations**  Besides diphone, triphone is another way to define context-dependent representations for phonemes. Each triphone class consists

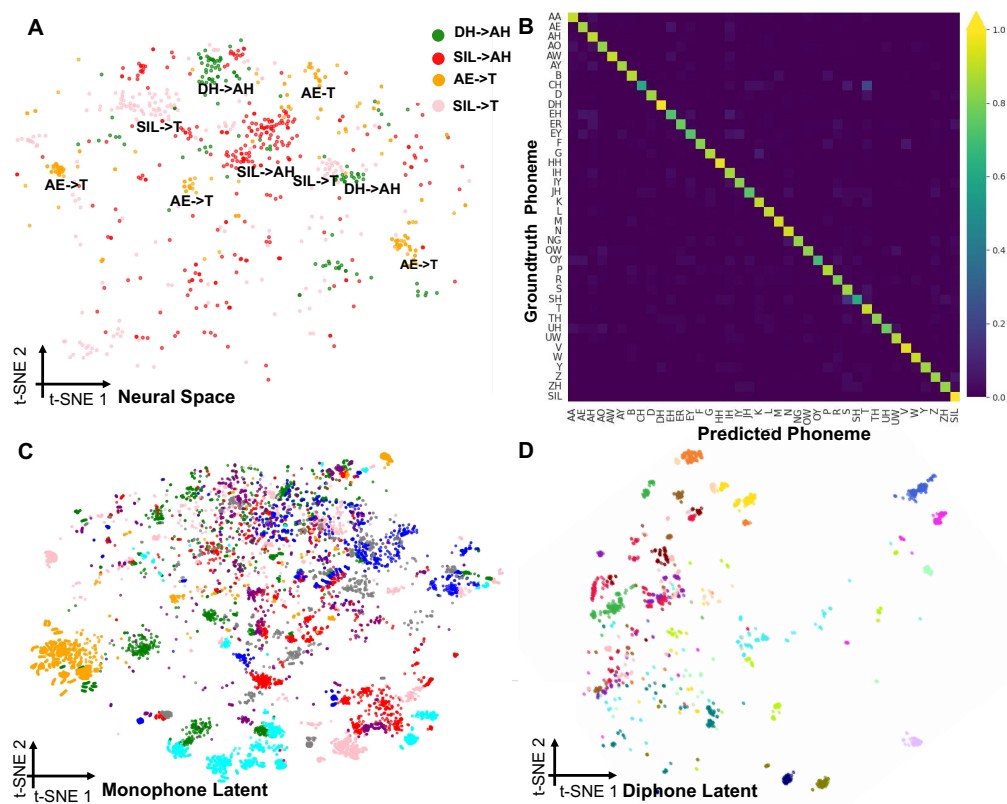

Figure 3: **A**: 2D t-SNE visualization of neural signal projections illustrating the context-dependent nature of phonemes in neural reprentations. Different colors indicate different diphone classes. **B**: Confusion matrix of ground truth phonemes vs. DCoND's predicted phonemes. **C**: 2D t-SNE visualization for the latent space of the neural decoder trained with single phoneme decoding objective (Monophone). Different colors indicate different phoneme classes. **D**: 2D t-SNE visualization for the latent space of the neural decoder trained with diphone decoding objective. Different colors indicate different diphone classes.

Table 3: Ablation study on alternative definitions of context-aware phoneme representations.

| | DCoND-L | Triphone | | | |
| | | K=50 | K=100 | K=200 | Grouping |
|---|---|---|---|---|---|
| PER×100 ↓ | 15.34 | 16.01 | **15.02** | 15.11 | 28.55 |
| WER×100 ↓ | **8.06** | 9.69 | 9.67 | 9.81 | 13.98 |

of three consecutive phonemes, e.g. $H \rightarrow OW \rightarrow P$, providing a finer granularity of context dependency with $40^3$ possible classes. Such a large number of classes can be overwhelming for the model to learn. Given that many of them have few to no presence in the data, to efficiently maintain a manageable size of decoding classes we select the top $K$ combinations of preceding and succeeding phonemes for each main phoneme, e.g. $* \rightarrow OW \rightarrow *$, based on their frequency of occurrence in the data, where $K \in [50, 100, 200]$. Alternatively, the preceding and succeeding phonemes could be grouped based on their articulatory similarity ("Grouping" in Table 3) (see Appendix A.2 for more details).

Results in Table 3 suggest that triphone with appropriate class size achieves comparable PER as the diphone counterpart (DCoND-L). However, triphone modeling underperforms diphone modeling in terms of WER, possibly because reducing the triphone's class size skews the phoneme distribution output of the neural decoder, making it incompatible with the distribution the subsequent 5-gram

model was originally trained on. Notably, the "grouping" method despite yielding a class size similar to that of $K = 200$, performs signficantly worse in both PER and WER. This implies that neural encoding for phonemes is more intricate, and grouping phonemes based on pronunciation similarity may not be optimal. Overall, we empirically find diphone, with its context-dependent nature and manageable class size, to be the most suitable modeling choice for this task and dataset.

**Contribution of LLMs** LLMs play an important role in translating phonemes into sentences. As detailed in Section 3, our LLM-based phoneme-to-text pipeline consists of three steps: (i) transcription generation (*5-gram*), (ii) transcription rescoring (*OPT*), (iii) error correction via ensembling with *ICL GPT3.5* or *finetuned GPT3.5*. We show in Figure 4 how each step of the LLM pipeline contributes to the overall WER. In particular, we consider the following variants of LLMs on top of DCoND: *5-gram+OPT* as used in NPTL (DCoND-L), *5-gram+OPT+ICL GPT3.5* with context length of

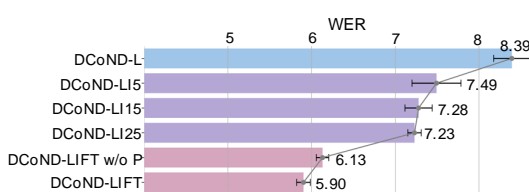

Figure 4: Ablation study on the contribution of LLMs.

5 (DCoND-LI5), context length of 15 (DCoND-LI15), and context length of 25 (DCoND-LI25), *5-gram+OPT+finetuned GPT3.5* without phoneme inputs (DCoND-LIFT w/o P), and our most performant model – *5-gram+OPT+finetuned GPT3.5* with phoneme inputs (DCoND-LIFT). We show that using GPT3.5 to refine the transcriptions from an ensemble of candidates, selected based on the highest re-scored likelihood given by the *5-gram+OPT* step, leads to an improvement in WER. Specifically, when GPT3.5 is exposed to ICL exemplars (DCoND-LI), its performance further improves as more exemplars are provided. However, finetuned GPT3.5 – unaffected by the limited ICL context length – enjoys more improvements in WER. The best WER is achieved when GPT3.5 leverages the predicted phonemes to refine the query transcriptions (DCoND-LIFT). Additional ablations are provided in Section A.3 of the Appendix.

## 5 DISCUSSION

In this work, we propose a divide-and-conquer approach for neural decoders (DCoND) together with an LLM-enhanced ensembling method (LI and LIFT) for decoding speech from neural activity. Motivated by a neuroscientific insight (coarticulation), DCoND leverages diphone, a context-dependent representation for phoneme sequences, as the modeling target. We show that decomposing the phoneme classification task into diphone classsfication subtasks facilitates the phoneme decoding task, subsequently improve the final sentence decoding accuracy. LI and LIFT propose an LLM-based ensembling approach where both phoneme sequence candidates and transcription candidates are provided as inputs to GPT3.5 to enhance its ability to refine the transcription candidates. We show that DCoND-LIFT achieves SOTA PER and WER on the Brain-to-Text 2024 Benchmark, outperforming leading methods by a large margin.

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

Table 4: Sensitivity analysis on Brain-to-Text 2024 Benchmark

|  | PER$\times 100 \downarrow$ | WER$\times 100 \downarrow$ | P-WER$\times 100 \downarrow$ |
|---|---|---|---|
| NPTL [46] | 16.62 | 9.46 | 11.33 |
| LISA [2] | – | 8.93 | – |
| DCoND-L (Ours) | $\mathbf{15.44 \pm 0.46}$ | $8.39 \pm 0.22$ | $\mathbf{8.09 \pm 1.62}$ |
| DCoND-LI (Ours) | – | $7.23 \pm 0.08$ | – |
| DCoND-LIFT (Ours) | – | $\mathbf{5.90 \pm 0.08}$ | – |

# A  APPENDIX

## A.1  SENSITIVITY ANALYSIS

We report the mean and standard deviation of DCoND-L, DCoND-LI and DCoND-LIFT in Table 4. The mean and standard deviation are obtained across 5 random seeds. The proposed methods (DCoND-L, DCoND-LI and DCoND-LIFT) maintain a significant gap over the NPTL and LISA baselines (Willett et al., 2023a; Benster et al., 2024).

## A.2  TRIPHONE AS AN ALTERNATIVE FOR CONTEXT-DEPENDENT PHONEME REPRESENTATION

Triphones expand upon diphones by incorporating a larger context. Specifically, a triphone considers one phoneme before and one phoneme after the current main phoneme. Consequently, when a neural signal segment is decoded into acoustic units based on the continuity of three phonemes, it reflects a triphone structure. For example, the single phoneme sequence

$$H, \quad OW, \quad P$$

for "hope", can be transferred to triphone

$$\text{``}SIL \to H \to OW, \quad H \to OW \to P, \quad OW \to P \to SIL\text{''}.$$

In this scenario, the time steps required for decoding single phonemes and triphones remain the same. However, triphones introduce a substantial increase in the number of classes, scaling as $N^3$, which can be prohibitively large (e.g., 64000 when $N = 40$). The divide and conquer idea in this case could be expressed as:

$$f(x) = p(Z = c_i | X) = \sum_{c_j \in C, c_q \in C} p(c_j, c_i, c_q | X)$$

Similar to the diphone probability matrix, these triphone classes are then mapped into a triphone matrix, where each element represents the probability of the current neural signal encoding the phoneme transition from phoneme $c_j$ to phoneme $c_i$ and concluding at phoneme $c_q$. By summing over the first and last dimensions, we obtain $p(Z = c_i | X)$. Given the potential sparsity of triphone combinations, certain triphone subclasses may not occur frequently in a given language. To mitigate this, we select the top $K$ subclasses for each triphone sample, based on occurrence counts within the current vocabulary. Specifically, for a main phoneme $c_i$, we rank all possible combinations of $* - > c_i - > *$ and retain the top $K$ as subclasses for the phoneme class $c_i$.

Additionally, aside from selecting the top $K$ subclasses, an alternative approach involves grouping phones according to articulation similarity Herff et al. (2015). This categorization leads to subclasses of the phoneme $c_i$ as $group_j - > c_i - > group_q$. We categorize phonemes into 14 groups, encompassing Bilabial Sounds, Labiodental Sounds, Dental Sounds, Alveolar Sounds, Palatal Sounds, Velar Sounds, Glottal Sounds, Front Vowels, Central Vowels, Back Vowels, and SIL. In this context, the number of subclasses amounts to $14 * 40 * 14$, which is comparable to the number of classes when $K = 200$ (resulting in a total of 200*40 subclasses).

## A.3  ADDITIONAL ABLATION STUDY ON THE CONTRIBUTION OF LMS

We conduct additional study to assess the role of phoneme-to-transcription generation and re-scoring methods (Figure 5). We show that removing the re-scoring step performed by the OPT model

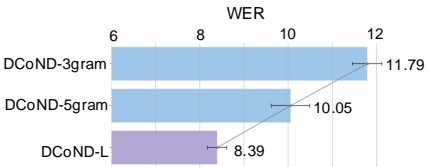

Figure 5: Ablation study on the contribution of re-scoring step in the phoneme-to-transcription pipeline.

Table 5: GPT-3.5 vs Llama-3.1-70B for error correction from ensemble of transcriptions

|  | Llama-3.1-70B WER | GPT 3.5 WER |
|---|---|---|
| DCoND-LI | 7.38 | 7.29 |
| DCoND-LIFT | 6.85 | 5.77 |

in DCoND-L significantly degrades WER (DCoND-3gram and DCoND-5gram), highlighting the importance of the transcription re-scoring step. In addition, the 5-gram model with longer phoneme dependency generates more accurate transcription candidates compared to the 3-gram model.

### A.4 OPEN-SOURCE LLMS FOR DCOND-LI & DCOND-LIFT

In addition to the closed-source GPT-3.5, we explore the use of the open-source Llama-3.1-70B for refining transcription predictions. We evaluated Llama-3.1-70B in both in-context learning (DCoND-LI) and fine-tuning (DCoND-LIFT) scenarios and compare it against GPT3.5 (Table 5). Llama-3.1-70B performs on par with GPT3.5 in ICL setting, while closely trail behind in finetuning setting, all the while outperforming NPTL and LISA baselines. These results demonstrate our method's robustness and generalizability to other LLMs besides GPT3.5, and warrant the accessibility of our methods to the broad community.

### A.5 INVESTIGATION ON ARCHITECTURE CHOICES FOR NEURAL DECODERS

We study the effects of different model architectures on the phoneme decoding performance (PER) (Table 6). We observe a significant performance degradation in PER when using Transformer as the neural decoder. On the other hand, RNN counterparts (LSTM and GRU) perform decently well, with GRU being the most performant model for both single phoneme decoding (NPTL) and diphone decoding (DCoND).

### A.6 IMPLEMENTATION DETAILS

We preprocess the neural signal and construct an RNN neural encoder following the methodology outlined in Willett et al. (2023a). The raw neural signal $X \in \mathbb{R}^{T \times D}$ is initially partitioned into smaller patches with a window size of $W$, resulting in a patched neural signal of shape $X \in \mathbb{R}^{T' \times (DW)}$. Overlapping between patches is permitted and determined by the stride size. $W = 14$ for diphone experiments and 32 for the triphone experiemnts. The bidirectional RNN processes these patched neural signals as inputs, which are subsequently transformed into the neural representation space $H = [h_1, h_2, \cdots, h_{T'}] \in \mathbb{R}^{T' \times d}$. A fully connected layer then maps the hidden representations to

Table 6: Comparison of different model architectures on phoneme decoding performance

|  | PER | | |
|---|---|---|---|
|  | Transformer | LSTM | GRU |
| NPTL | $39.58 \pm 0.15$ | $17.49 \pm 0.32$ | $16.63 \pm 0.19$ |
| DCoND | $38.88 \pm 0.17$ | $16.08 \pm 0.23$ | $15.44 \pm 0.46$ |

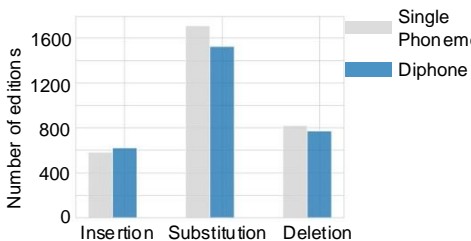

Figure 6: Phoneme error types analysis during single phoneme decoding and diphone.

diphone or triphone subclasses, denoted as $P(S = s_i|X)$. The outputs of the fully connected layer are used to compute $\mathcal{L}_s$. The computation of single phoneme probabilities is detailed in Equation 3. We merge the probability computed from diphone or triphone.

During the RNN training, we utilize a batch size of 32, a learning rate of 0.02, and the Adam optimizer across various experiments the same set of parameters as used in NPTL baseline Willett et al. (2023a). To facilitate diphone and triphone learning, we initially train the subclasses for 10 epochs and then gradually increase the ratio of the single phoneme loss by 0.1 every 10 epochs until it reaches 0.6. The number of training epochs varies for single phoneme learning, diphone learning, and triphone learning. Specifically, we conduct experiments for up to 100 epochs for single phoneme learning (NPTL baseline), 120 epochs for diphone learning, and 140 epochs for triphone learning since the diphone and triphone required additional subclass training procedures. Increasing the number of training epochs can often lead the model to overfit the training data. Training was done on 2 GeForce RTX 2080 Ti with around 12GB memory. The training take around 6-8 hours.

The 5-gram model takes the predicted phoneme logits as inputs, which can be scaled by a temperature factor denoted as $t$ using the formula $logits := logits/t$. Through experimentation, we have found that setting $t = 1.2$ generally improves the decoding performance. Therefore, we use $t = 1.2$ for our experiments, including the implementation of NPTL, which has resulted in improved baseline results. Specifically, the leaderboard score has improved from 9.76 to 9.46.

### A.7 PHONEME ERROR ANALYSIS

We conducted a detailed analysis of the various types of errors encountered during phoneme decoding. This analysis involved assessing the operations necessary to align the decoded phoneme sequence with the ground truth phonemes, comparing scenarios where only single phoneme decoding is used versus employing diphone subclass decoding. Overall, our findings indicate that employing diphone subclass decoding leads to a reduction in the number of operations required to align the decoded sequence with the ground truth phonemes. Specifically, fewer editing operations, particularly substitutions, are needed when utilizing the diphone decoding paradigm compared to directly decoding single phonemes.

### A.8 PROMPT FOR GPT3.5

**Prompt to GPT3.5** : Your task is to perform automatic speech recognition. Below are multiple candidate transcriptions together with their corresponding phoneme representations. The phonemes are taken from the CMU Pronouncing Dictionary. The special symbol SIL represents the start of the sentence, or the end of the sentence, or the space between two adjacent words. Based on the transcription candidates and their phoneme representations, come up with a transcription and its corresponding phoneme representation that are most accurate, ensuring the transcription is contextually and grammatically correct. Focus on key differences in the candidates that change the meaning or correctness. Avoid selections with repetitive or nonsensical phrases. In cases of ambiguity, select the option that is most coherent and contextually sound, taking clues from the phoneme representations. The candidate phoneme representations may not always be the correct representation of the corresponding candidate transcriptions. Some phonemes in the candidate phoneme sequences might have been incorrectly added, removed, or replaced. However, the candidate phonemes contain useful information that will help you come up with the correct transcription and phoneme representation. You should translate each subgroup of phonemes that is enclosed by two SIL

symbols into one single word. You should remove SIL symbols at the start or the end of the phoneme sequence. Respond with your refined transcription and its corresponding phoneme representation only, without any introductory text.

**Examples of prediction and correction pairs**    Transcription candidate 1: but we don't know that. Transcription candidate 2: but we don't know that. Transcription candidate 3: but you don't know that. Transcription candidate 4: but you don't know that. Transcription candidate 5: but you don't know that. Transcription candidate 6: but you don't know that. Transcription candidate 7: but you don't know that. Transcription candidate 8: but you don't know that. Transcription candidate 9: but we don't know that. Transcription candidate 10: but we don't know that. Phoneme candidate 1: SIL B AH T SIL W IY SIL D OW N T SIL N OW SIL DH AE T SIL. Phoneme candidate 2: SIL B AH T SIL Y IY SIL D OW N T SIL N OW SIL DH AE T SIL. Phoneme candidate 3: SIL B AH T SIL Y UW SIL D OW N T SIL N OW SIL AE T SIL. Phoneme candidate 4: SIL B AH T SIL Y UW SIL D OW N T SIL N OW SIL DH AE T SIL. Phoneme candidate 5: SIL B AH T SIL DH UW SIL D OW N T SIL N OW SIL DH AE T SIL. Phoneme candidate 6: SIL B AH T SIL Y UW SIL D OW N T SIL N OW SIL DH AE T SIL. Phoneme candidate 7: SIL B AH T SIL Y UW SIL D OW N T SIL N OW SIL DH AE T SIL. Phoneme candidate 8: SIL B AH T SIL Y UW SIL D OW N T SIL N OW SIL DH AE T SIL. Phoneme candidate 9: SIL B AH T SIL W IY SIL D OW N T SIL N OW Z SIL DH AE T SIL. Phoneme candidate 10: SIL B AH T SIL DH IY SIL D OW N T SIL N OW SIL AE T SIL.

Table 7: Example of In-Context-Learning (ICL) prompts and query.

| | |
|---|---|
| **System Prompt:** | Your task is to perform automatic speech recognition. You are given ten candidates of an unknown transcription. Your job is to come up with a transcription that is most accurate, relying on the context that the candidates provide. First, observe the provided examples demonstrating how the task should be done, then work on the query candidates. In each example, ten transcription candidates, their corresponding phoeneme representations, and a ground truth transcription are given. The ground truth transcription is the correct transcription, while the transcription candidates and phoneme representations may or may not contain errors. Some phonemes in the phoneme sequences might have been incorrectly added, removed, or replaced. However, the phonemes contain helpful information that will help you come up with the correct transcription. You should translate each subgroup of phonemes that is enclosed by two SIL symbols into one single word. You should remove SIL symbols at the start and the end of the phoneme sequence. Make sure your transcription based on the query candidates is contextually and grammatically correct. Focus on key differences in the candidates that change the meaning or correctness. Avoid selections with repetitive or nonsensical phrases. In cases of ambiguity, select the option that is most coherent and contextually sound. Respond with your final transcription only, without any introductory text. |
| **Context prompt:** | **Example 1**: *Transcription candidate 1*: i enjoyed it very much. ⋯ *Transcription candidate 10:* i enjoyed it very much. *Phoneme candidate 1:* AY SIL EH N JH OY D SIL IH T SIL V EH R IY SIL M AH CH SIL. ⋯ *Phoneme candidate 10:* AY SIL EH N JH OY D SIL IH T SIL V EH R IY SIL M AH CH SIL. ⋯ **Ground truth phonemes:** AY SIL EH N JH OY D SIL IH T SIL V EH R IY SIL M AH CH. **Ground truth transcription**: i enjoyed it very much. ⋯ **Example N:** *Transcription candidate 1:* the ranks of asian riders are falling too. ⋯ *Transcription candidate candidate 10:* the ranks of asian riders are willing to. *Phoneme candidate 1*: DH AH SIL R AE NG K S SIL AH V SIL EY ZH AH N SIL R AY D Z SIL AA R SIL F L D IH NG SIL T UW SIL. ⋯ *Phoneme candidate 10*: DH AH SIL R AE K S SIL AH V SIL EY ZH AH N SIL R EY D ER Z SIL AA R SIL F IY L IH NG SIL T UW SIL. **Ground truth phonemes**: DH AH SIL R AE NG K S SIL AH V SIL EY ZH AH N SIL R AY D ER Z SIL AA R SIL S W EH L IH NG SIL T UW. **Ground truth transcription**: the ranks of asian riders are swelling too |
| **Query:** | Transcription candidate 1: i'm originally from colorado. ⋯ Transcription candidate 10: i'm only from colorado. Phoneme candidate 1: SIL AY M SIL ER N AH L IY SIL F R AH M SIL K AO L ER AA D OW SIL. ⋯ Phoneme candidate 10: SIL AY M SIL AH N L IY SIL F R AH M SIL K AO L R AA D OW SIL. |

