# OpenReview forum: "Brain-to-Text Decoding with Context-Aware Neural Representations and Large Language Models"
_ICLR.cc/2025/Conference — ICLR 2025 Conference Withdrawn Submission_

### Official Review · Reviewer_aoXe · 2024-11-01

**Soundness:** 3
**Presentation:** 2
**Contribution:** 2
**Rating:** 3
**Confidence:** 2

**Summary:**

The paper presents a novel approach to decoding speech from neural activity, focusing on individuals with neurological conditions that impair their ability to articulate speech. The key contributions of the paper are:

1. DCoND (Divide-and-Conquer Neural Decoder): A new framework that decodes phonemes from neural activity during attempted speech. DCoND leverages the context-dependent nature of phonemes by using diphones (sequences of two adjacent phonemes) as the intermediate representation, which enhances phoneme decoding performance.

2. LLM-Enhanced Ensembling: The paper proposes incorporating decoded phonemes alongside decoded words in a Large Language Model (LLM)-based ensembling strategy to improve speech decoding performance. It also introduces the use of In-Context Learning (ICL) as an efficient alternative to fine-tuning LLMs, offering a more resource-efficient solution for brain-to-text systems.

3. State-of-the-Art Performance: The proposed DCoND-LIFT (DCoND with LLM FineTuning) achieves state-of-the-art (SOTA) performance on the Brain-to-Text 2024 benchmark, with a Phoneme Error Rate (PER) of 15.34% and a Word Error Rate (WER) of 5.77%, significantly outperforming the leading SOTA method with an 8.93% WER.  The paper includes extensive ablation studies and analyses to demonstrate the effectiveness of the proposed methods, including trade-offs between diphone and monophone losses, alternative context-dependent phoneme representations, and the contribution of LLMs in the decoding pipeline.

Overall, the paper advances the field of brain-computer interfaces by offering a more accurate and efficient method for decoding speech from neural activity, with potential applications in restoring communication abilities for individuals with speech impairments.

**Strengths:**

The paper introduces a divide-and-conquer strategy (DCoND) that leverages diphones, a context-dependent representation of phonemes, to enhance the decoding performance. This approach is grounded in neuroscientific insights about the variability and context-dependence of neural signals during speech production.

The paper demonstrates the effectiveness of the proposed methods on the Brain-to-Text 2024 benchmark, achieving state-of-the-art performance in terms of Phoneme Error Rate (PER) and Word Error Rate (WER). The paper also includes detailed ablation studies and sensitivity analyses to validate the contributions of different components of the proposed system.

**Weaknesses:**

While the paper presents a promising approach to decoding speech from neural activity, it primarily builds upon established methods from the field of speech recognition, particularly in the use of diphones and language models. The novelty of the paper appears to be limited to the application of these mature techniques from hybrid speech recognition to the specific context of brain-to-text decoding, rather than introducing fundamentally new methodologies or insights.

Given the nature of the paper and its focus on applying existing methods from speech recognition to the brain-to-text decoding domain, it might be more appropriate for the authors to submit their work to journals that specialize in interdisciplinary research or those that focus on practical applications of existing technologies.

From the perspective of academic paper writing, I recommend that the authors elaborate further on the Divide-and-Conquer strategy and its innovative aspects compared to traditional speech recognition methods. Expanding on this topic will not only enhance the depth of the paper but also assist readers in better comprehending the unique characteristics and advantages of this approach. Such a discussion will more effectively highlight the novel contributions of the research and provide valuable insights and references for scholars in related fields.

**Questions:**

Could you briefly explain the main technical differences between speech-to-text and brain-to-text tasks?

---

> ### Author Response · Authors · 2024-11-23
>
> We thank the reviewer for their valuable feedback. We appreciate the reviewer comments that our proposed method is grounded in neuroscientific insights and achieves state-of-the-art performance in the Brain-to-Text decoding benchmark. We have addressed all the raised concerns below and revised the manuscript accordingly.
>
> >**W1**: re: novelty of our methods
>
> 1. We would like to emphasize that DCoND is not simply an application of the diphone idea from the ASR community. Unlike existing ASR models, DCoND does not simply decode diphones - it uses diphone decoding as an intermediate step to decode monophones. The main task of modeling monophone probabilities is broken down into subtasks of modeling diphone probabilities, after which the marginalization over diphones is done to obtain the monophone probabilities. **This newly introduced marginalization step, which integrates diphone and monophone decoding in a unified framework, represents the primary novelty that sets DCoND apart from traditional ASR methods.**
>
>    We provide additional experiments to demonstrate the effectiveness of collaboratively decoding diphones and monophones through marginalization (as in DCoND) vs. decoding diphones alone (as in ASR methods). Indeed, empirical results below demonstrate that decoding monophone with diphone marginalization (DCoND) resulted in a better PER than decoding using diphones alone. When coupled with a 3-gram language model that transcribes monophone sequences to word sequences, DCoND also yields a better WER than a diphone model followed by a 3-gram language model designed for diphone-to-word transcription.
>
>     |      | Diphone       | DCoND         |
>     |------|---------------|---------------|
>     | PER  | 19.14 $\pm$0.08 | 15.44$\pm$0.46 |
>     | WER  | 12.73$\pm$0.22  | 11.79 $\pm$0.30  |
>
>     The superior performance of DCoND can be attributed to its marginalization formulation, which allows for more imperfect diphone predictions - as long as the marginalized probability attains the highest value at the correct monophone, the model does not need to assign the highest probability at the correct diphone. When the amount of paired neural-text data is insufficient to train a good diphone decoding model - as is the case in the neuroscience domain - the marginalization step in DCoND proves to be an effective strategy.
>
> 2. Additional novelty in our work is in the non-trivial application of **context-dependent representations in the neuroscience domain**. To the best of our knowledge, no previous studies have demonstrated that neural activity underlying attempted speech contains and can be decoded into context-dependent representations, let alone leveraging them to enhance performance of brain-to-text decoding models. DCoND-L with WER of 8.06 outperforms NPTL with an 14.80% improvement, proving the significance of using context-dependent representations for neural signals. Our novelty on this front provides both scientific insights and algorithmic solutions to the brain-to-text decoding problem, bringing significant values for the neuroscience community at ICLR.
>
> 3. Our contributions in the phonemes-to-words pipeline involve enhancements particularly geared towards the **data-limited regime of neuroscience applications via the efficient use of existing available resources**:
>
>     *(i)* combining multiple LLMs working as a multi-agent system, each specialized for distinct steps in the word decoding pipeline, to improve the overall word decoding accuracy.
>
>    *(ii)* incorporating decoded phonemes as an informative source of reference for language models to reduce the error accumulation in the multi-agent system.
>
>     *(iii)* demonstrating the efficiency of in-context learning paradigm for rapid inference in resource-constrained settings.
>
>     These enhancements together efficiently and significantly improve WER of the brain-to-text pipeline, with 18.4% and 35.4% improvements over the best baseline for DCoND-LI and DCoND-LIFT, respectively.
>
> **We will summarize and outline these novelties in the revised version of the paper.**
>
> > **W2**: re: relevance to the ICLR community
>
> As we elaborated in our response above, our paper offers unique technical novelties and scientific insights, not simply a trivial application of existing technologies. We submitted our paper under the “Applications to neuroscience & cognitive science” subject area of ICLR, as we believe our contributions offer technical novelties that bridge neuroscience and machine learning. These values align closely with the ICLR community’s interest in advancing interdisciplinary research.

---

> ### Author Response · Authors · 2024-11-23
>
> > **W3**: re: elaboration of the Divide-and-Conquer method
>
> We thank the reviewer for the suggestion to provide further details regarding our Divide-and-Conquer strategy. We included the empirical results (please see further description in response to W1) and additional intuitive explanation on why DCoND is better than a model that decodes diphones only. We will revise the manuscript further to include a primer on the Divide-and-Conquer method upon preparing the final version of the manuscript.
>
> > **Q1**: re: technical differences between brain-to-text and speech-to-text
>
> Our technical differences lie in the newly developed DCoND algorithm for brain-to-phonemes neural decoder as well as the multi-agent LLM system for phoneme-to-text transcription. These components are unique and critical for brain-to-text decoding and have not appeared in speech-to-text context. We provided more information on these in our response to W1 and the original manuscript (please see ‘Brain-to-Text Decoding vs. Speech-to-Text Decoding’ in the Related Work section). We will make sure to include further discussion in related work upon preparing the final version of the manuscript.

---

### Official Review · Reviewer_QvtA · 2024-11-01

**Soundness:** 3
**Presentation:** 3
**Contribution:** 2
**Rating:** 5
**Confidence:** 4

**Summary:**

This paper tackles the challenge of converting brain activity to text as part of Brain-Computer Interfaces for individuals with speech impairments. The primary contributions of the paper are:

- The use of diphones, rather than monophones, for mapping neural activity to targets. Previous work has relied on monophones, which can be noisy since phones are influenced by preceding sounds during articulation.

- The application of an LLM with in-context learning examples for error correction.

Experimental results demonstrate the effectiveness of the proposed approach, achieving state-of-the-art performance on the Brain-to-Text benchmark.

**Strengths:**

- The approach of using diphones and subsequently normalizing them to monophones is well-motivated and delivers superior performance.

- The experimental results, where LLMs are employed for error correction, show significant improvements over strong baselines.

- The ablation study, which explores different architectures (including transformers), open-source LLMs, and various in-context learning examples for LLMs, thoroughly examines multiple aspects of the approach.

**Weaknesses:**

- While the results are state-of-the-art, the major contributions of this paper are established concepts within the ASR literature. The use of diphones and triphones has been common in non-neural methods to address articulation issues.

- The application of LLMs for ASR correction is also a well-explored area in the field.

**Questions:**

1. In the paper, diphones were normalized to monophones before being used with an n-gram language model to obtain word sequences. Is this normalization necessary? An n-gram language model could also be trained to directly transform diphone sequences into word sequences.

2. What effect would using the richer context of diphone sequences have when processing with LLMs, as opposed to using the normalized monophone sequences?

---

> ### Author Response · Authors · 2024-11-23
>
> We thank the reviewer for their insightful and valuable feedback. We appreciate the reviewer's comments that our proposed method is well-motivated and delivers superior performance over strong baselines with thorough experiments. We addressed all the concerns of the reviewer below and revised the manuscript accordingly.
>
> > **W1**: re: novelty of our methods
>
> We address the first and second weakness below by clarifying the novelty of our work, which includes the technical contributions of the proposed methods and valuable insights for the neuroscience domain.
>
> 1. We would like to emphasize that DCoND is not simply an application of the diphone concept from the ASR community. Unlike existing ASR models, DCoND does not simply decode diphones - it uses diphone decoding as an intermediate step to decode monophones. The main task of modeling monophone probabilities is broken down into subtasks of modeling diphone probabilities, after which the marginalization over diphones is done to obtain the monophone probabilities. **This newly introduced marginalization step, which integrates diphone and monophone decoding in a unified framework, represents the primary novelty that sets DCoND apart from traditional ASR methods**.
>
>
> 2. Additional novelty in our work is in the non-trivial application of **context-dependent representations in the neuroscience domain**. To the best of our knowledge, no previous studies have demonstrated that neural activity underlying attempted speech contains and can be decoded into context-dependent representations, let alone leveraging them to enhance performance of brain-to-text decoding models. DCoND-L with WER of 8.06 outperforms NPTL with an 14.80% improvement, proving the significance of using context-dependent representations for neural signals. Our novelty on this front provides both scientific insights and algorithmic solutions to the brain-to-text decoding problem, bringing significant values for the neuroscience community at ICLR.
>
>
> **We will summarize and outline these novelties in the revised version of the paper.**
>
> > **W2**: re: novelty regarding our use of LLMs for error correction
>
> Although LLMs have been used for ASR correction, decoded phonemes are hardly used as references for correction. In our paper we demonstrate that decoded phonemes can also contribute to the accuracy of transcription error correction, where DCoND-LIFT with phoneme inputs reaches an average WER of 5.9 vs. 6.13 without phoneme inputs.
> We would like to re-emphasize that our contributions on the transcription error correction include **enhancements particularly geared towards the data-limited regime of neuroscience applications via the efficient use of existing available resources**:
> * (i) incorporating decoded phonemes as an informative source of reference for language models to reduce the error accumulation induced during the use of LLMs for error correction.
> * (ii) combining multiple LLMs working as a multi-agent system, each specialized for distinct steps in the word decoding pipeline, to improve the overall word decoding accuracy.
> * (iii) demonstrating the efficiency of in-context learning paradigm for rapid inference in resource-constrained settings.
> These enhancements together efficiently and significantly improve WER of the brain-to-text pipeline, with 18.4% and 35.4% improvements over the best baseline for DCoND-LI and DCoND-LIFT, respectively.
>
> > **Q1&Q2**: re: benefits of the marginalization step
>
> We appreciate the reviewer’s suggestion for additional analyses regarding the benefits of the marginalization step. Below we provide additional findings to demonstrate the effectiveness of collaboratively decoding diphones and monophones through marginalization (as in DCoND) vs. decoding diphones alone (as in ASR methods). Indeed, empirical results demonstrate that decoding monophone with diphone marginalization (DCoND) resulted in a better PER than decoding using diphones alone. When coupled with a 3-gram language model that transcribes monophone sequences to word sequences, DCoND also yields a better WER than a diphone model followed by a 3-gram language model designed for diphone-to-word transcription.
> |      | Diphone       | DCoND         |
> |------|---------------|---------------|
> | PER  | 19.14$\pm$0.08 | 15.44$\pm$0.46 |
> | WER  | 12.73$\pm$0.22  | 11.79$\pm$0.30  |
>
> The superior performance of DCoND can be attributed to its marginalization formulation, which allows for more imperfect diphone predictions - as long as the marginalized probability attains the highest value at the correct monophone, the model does not need to assign the highest probability at the correct diphone. When the amount of paired neural-text data is insufficient to train a good diphone decoding model - as is the case in the neuroscience domain - the marginalization step in DCoND proves to be an effective strategy.

---

### Official Review · Reviewer_y5i9 · 2024-11-05

**Soundness:** 2
**Presentation:** 2
**Contribution:** 2
**Rating:** 3
**Confidence:** 4

**Summary:**

This paper focuses on decoding attempted speech from neural activity, a technology that can aid people with certain disabilities. Instead of representing the output with phonemes as done by other studies, they propose to represent the output with diphones, which can better model the coarticulation effects. They propose a divide-and-conquer strategy for decoding, which marginalizes the biphone distribution to get phoneme distribution. They also propose utilizing LLMs to improve the results, and in particular, an in-context learning paradigm as compared to fine-tuning methods. They report SotA results on the brain-to-text 2024 benchmark.

**Strengths:**

The paper is mostly well-written and easy to follow (with some exceptions of the technical aspect which I will mention in the weakness section). The problem it works on is a potentially important one, which leverages technology to help those with speech impairments, and the findings can potentially be beneficial for other fields where neural activity is used. They report SOTA results in the benchmark used, and present interesting analysis delving into the inner workings of the model.

**Weaknesses:**

Although the paper does a generally good job describing methods with textual descriptions, there are a few issues when it comes to mathematical notations. Some examples are,
- in Section 3, in the problem statement, the authors use Z to represent phoneme sequence, with Z being a vector of R (real numbers). This is an unusual choice since Z is discrete representing different identities of phonemes.
- In the same paragraph, the author mentioned using GRU to model the function f: X -> Z. When I read this, I was confused since X and Z are typically of different lengths, involving a hidden alignment between x and z, which is beyond the modeling capacity of GRUs. The authors, in a later section, mentioned using a CTC model to learn the implicit alignment. It would be better to mention that first in the method description section to reduce confusion and misunderstanding.
- I am having a bit of trouble understanding Equation (1). From the previous definition, Z and X are both multi-dimensional vectors, but here, if I understand correctly, X and Z are single-dimensional numbers. Some clarification on this would be beneficial.
- The authors mentioned the issue of sparsity when discussing triphones, but I don't see explicit mention of how they deal with diphones. With just 40 phonemes, there are 1600 different possible diphones, and I would imagine the distribution among those 1600 choices can be highly unbalanced. It would be helpful to see stats regarding sparsity in those combinations. In the earlier days of speech recognition research where modular models were used, a common way to deal with sparsity was to build phonetic decision trees to cluster different triphones into equivalent classes. I would imagine this would also be a good choice to make triphone work in the context of this paper. So I'm not fully convinced that diphones would outperform triphones, if using clustering techniques like decision trees.
- It seems that in generating the representation of output, the authors assume there is always going to be SIL between words. I'm not sure if this is a reasonable assumption without providing justification or comparisons.

In section 4.3, there are references to Table 7 which I believe is a typo. Should be Table 1?

**Questions:**

Please see weakness.

---

> ### Author Response · Authors · 2024-11-23
>
> We thank the reviewer for their insightful and valuable feedback. We appreciate the reviewer’s comments that our proposed method may be beneficial for other fields and includes interesting analyses. We have addressed all the raised concerns below and revised the manuscript accordingly, including updates to the notations and technical descriptions.
>
> > **W1**: re: notation for Z
>
> We thank the reviewer for pointing out the issue in the current definition of Z. We have revised the definition to $Z \in \mathbb{Z}^{T’}$, to indicate that Z takes discrete values indicating phoneme classes.
>
> > **W2**: re: earlier introduction of CTC loss in the text
>
> We appreciate the reviewer’s suggestion of mentioning the CTC first in the method description to improve the clarity of the text. We have revised the paper to mention the CTC loss when introducing the GRU model.
>
> > **W3**: re: notations for dimensions of X and Z
>
> We thank the reviewer for their suggestion to clarify the dimensions of the variables. X is a two dimensional matrix ($X \in R ^{T \times D}$), $Z$ is a one dimensional vector with length $T’$ as introduced in the Problem Formulation. For simplicity, equation 1 only shows the prediction for the monophone $Z$ at a single timestep, i.e. $T’ = 1$. We have added clarification for the dimensionality in the revised manuscript as the reviewer suggested.
>
> >  **W4**: re: how our method handles sparsity issues of diphones and triphones
>
> The sparsity (percentage of diphone/triphone classes that don’t exist in the training set) is 36.97% for diphones and 88.72% for triphones. The sparsity is addressed for both diphones and triphones by our proposed marginalization method. For the diphone case, the 1600 diphone classes are reduced to 40 monophone classes by the marginalization process, where the monophone loss on these 40 classes are computed. This incorporation of monophone loss beside the diphone loss helps the DCoND model tolerate some errors on the diphone predictions, making DCoND less susceptible to the sparsity issue brought about by the use of diphones.
>
> To deal with sparsity issues in triphones, in Table 3 we have shown the performance of a clustering method which shares a similar idea to the phonetic decision tree clustering method the reviewer suggested. We clustered 40 phonemes into 14 classes based on articulation similarity, as proposed by Herff et al. [1]. Using these groupings, we constructed triphones with the structure: $cluster\_j \rightarrow phoneme\_i \rightarrow cluster_q$,  where there are 14, 40, and 14 classes for $cluster_j$, $
> phoneme_i$, $cluster_q$, respectively. We included more details in Appendix A.2. We showed in Table 3 that this grouping approach significantly underperformed DCoND, with PER and WER being 28.55 and 13.98, respectively, significantly worse than 15.34 PER and 8.06 WER of DCoND.
>
> > **W5**: re: the presence of SIL between words
>
> The dataset used in our study was collected from a patient with Amyotrophic Lateral Sclerosis (ALS). The patient, due to their medical condition, is unable to speak swiftly. Therefore the produced speech contains noticeable SIL between words. Prior works have demonstrated that including SIL as a decoding target improves overall decoding performance compared to models that do not explicitly account for SIL [2]. These findings provide a baseline for our approach to support the inclusion of SIL in generating the representation of output.
>
>
> >  **W6**: re: typo in section 4.3
>
> We thank the reviewer for pointing out the typo. We have changed the reference to Table 1 in the revised manuscript.
>
>
> [1] Christian Herff, Dominic Heger, Adriana De Pesters, Dominic Telaar, Peter Brunner, Gerwin Schalk, and Tanja Schultz. Brain-to-text: decoding spoken phrases from phone representations in the brain. Frontiers in neuroscience, 9:217, 2015.
>
> [2] Willett, Francis R., Erin M. Kunz, Chaofei Fan, Donald T. Avansino, Guy H. Wilson, Eun Young Choi, Foram Kamdar et al. "A high-performance speech neuroprosthesis." Nature 620, no. 7976 (2023): 1031-1036.

---

### Official Review · Reviewer_4p8d · 2024-11-07

**Soundness:** 3
**Presentation:** 3
**Contribution:** 2
**Rating:** 5
**Confidence:** 3

**Summary:**

This work proposes a multi-stage brain-to-text decoding framework consisting of an RNN phone decoding model integrating diphone as context-aware representations, and an LLM-based ensembling strategy utilizing both the decoded words by n-gram models and decoded phonemes as input to an fine-tuned LLM or by the ICL manner for correction. They demonstrate the effectiveness on the Brain-to-Text 2024 benchmark, where their approach achieves state-of-the-art (SOTA) PER of 15.34% and WER of 5.77%.

**Strengths:**

1. This paper is generally well written and easy to follow. The authors illustrate their methodology in a well-organized way and the results on Brain-to-Text 2024 are strong to prove its effectiveness.

2. Brain-to-text decoding could offer promising pathways for restoring communication ability and thus is a research topic that can make difference.

**Weaknesses:**

1. Utilizing diphone or triphone is a well-known technique for ASR community for enhancing phoneme decoding, and application to brain-to-text decoding shows some merits, but lacks some novelty in general. Also, the authors improve the ensembling strategy by integrating both the phoneme and word sequences for LLM, which is also a minor modification of the ensembling pipeline.

2. It seems to me the phoneme to sequence pipeline is a combination of NPTL and LISA besides the integration of the phoneme information. The procedure is a bit lengthy and there could be better way for improvement considering both efficiency and effectveness.

**Questions:**

1. If I understand it correctly, the authors utilize similar 5-layer RNN for phoneme decoding as in NPTL? I suggest the authors provide detailed specifications of their models in the experimental parts.

2. The authors analyze the tradeoff between diphone and monophone losses in Table 2, I wonder if the hypeparameter is generalizable for different datasets.

---

> ### Author Response · Authors · 2024-11-23
>
> We thank the reviewer for their insightful and valuable feedback. We appreciate the reviewer’s comments that our paper is well-organized, providing strong results and can make a difference. We address the reviewer's concerns below regarding novelties and technical contributions.
>
> >**W1**: re: novelty of our methods
>
> 1. We would like to emphasize that DCoND is not simply an application of the diphone idea from the ASR community. Unlike existing ASR models, DCoND does not simply decode diphones - it uses diphone decoding as an intermediate step to decode monophones. The main task of modeling monophone probabilities is broken down into subtasks of modeling diphone probabilities, after which the marginalization over diphones is done to obtain the monophone probabilities. **This newly introduced marginalization step, which integrates diphone and monophone decoding in a unified framework, represents the primary novelty that sets DCoND apart from traditional ASR methods.
>
>     We provide additional experiments to demonstrate the effectiveness of collaboratively decoding diphones and monophones through marginalization (as in DCoND) vs. decoding diphones alone (as in ASR methods). Indeed, empirical results below demonstrate that decoding monophone with diphone marginalization (DCoND) resulted in a better PER than decoding using diphones alone. When coupled with a 3-gram language model that transcribes monophone sequences to word sequences, DCoND also yields a better WER than a diphone model followed by a 3-gram language model designed for diphone-to-word transcription.
>
>     |      | Diphone       | DCoND         |
>     |------|---------------|---------------|
>     | PER  | 19.14$\pm$0.08 | 15.44$\pm$0.46 |
>     | WER  | 12.73$\pm$0.22  | 11.79$\pm$0.30  |
>
>     The superior performance of DCoND can be attributed to its marginalization formulation, which allows for more imperfect diphone predictions - as long as the marginalized probability attains the highest value at the correct monophone, the model does not need to assign the highest probability at the correct diphone. When the amount of paired neural-text data is insufficient to train a good diphone decoding model - as is the case in the neuroscience domain - the marginalization step in DCoND proves to be an effective strategy.
>
> 2. Additional novelty in our work is in the non-trivial application of **context-dependent representations in the neuroscience domain**. To the best of our knowledge, no previous studies have demonstrated that neural activity underlying attempted speech contains and can be decoded into context-dependent representations, let alone leveraging them to enhance performance of brain-to-text decoding models. DCoND-L with WER of 8.06 outperforms NPTL with an 14.80% improvement, proving the significance of using context-dependent representations for neural signals. Our novelty on this front provides both scientific insights and algorithmic solutions to the brain-to-text decoding problem, bringing significant values for the neuroscience community at ICLR.
>
> 3. Our contributions in the phonemes-to-words pipeline involve enhancements particularly geared towards the **data-limited regime of neuroscience applications via the efficient use of existing available resources**:
>
>    *(i)* combining multiple LLMs working as a multi-agent system, each specialized for distinct steps in the word decoding pipeline, to improve the overall word decoding accuracy.
>
>    *(ii)* incorporating decoded phonemes as an informative source of reference for language models to reduce the error accumulation in the multi-agent system.
>
>     *(iii)* demonstrating the efficiency of in-context learning paradigm for rapid inference in resource-constrained settings.
>
>     These enhancements together efficiently and significantly improve WER of the brain-to-text pipeline, with 18.4% and 35.4% improvements over the best baseline for DCoND-LI and DCoND-LIFT, respectively.
>
> **We will summarize and outline these novelties in the revised version of the paper.**

---

> ### Author Response · Authors · 2024-11-23
>
> > **W2**: re: efficiency and effectiveness of our LLM pipeline
>
> Indeed, the current phoneme-to-word pipeline, LIFT, involves multiple steps. However, each integrated language model (LM) plays distinct roles in the LIFT pipeline (transcription generation, re-scoring, and error correction) which collectively contributes to the overall performance. In our Ablation Study (‘Contribution of LLMs’ in Section 4.5 and Appendix A.3) we demonstrated with extensive experiments how removing one of these steps would degrade the overall WER.
>
> In particular, when only 5-gram LM is used, DCoND-5gram achieves WER of 10.05 vs. 5.77 if using the full pipeline (Figure 5); when GPT3.5 is removed (DCoND-L), the WER is 8.06 (Table 1). Additionally, we carried out an experiment where the re-scoring step was removed, and observed that WER increased from 5.77 to 7.15. These results indicate that each LM has its own distinct contribution in the overall decoding performance.  A more efficient way using a single LM/LLM to achieve the current performance can be challenging, but could be an interesting direction for future research.
>
> > **Q1**: re: architecture details
>
> Yes, the reported results in the paper are obtained with a 5-layer RNN (GRU) as the model architecture for DCoND. In addition to GRU, we also conducted experiments with Transformer and LSTM architectures and included detailed results in Table 6 (Appendix A.5). We would like to emphasize that the novelty of DCoND is not on the neural decoder architecture, but rather on the algorithmic level (decoding monophones through marginalization of diphones) which is applicable to various neural decoders. Indeed Table 6 in Appendix A.5 shows that DCoND marginalization method is model-agnostic and consistently outperforms the monophone decoding baseline across all architecture designs, demonstrating its **robustness and versatility in accommodating diverse architectural choices**.
>
> We appreciate the reviewer’s suggestion to move the implementation details to the experimental section. We have included the implementation details in Appendix A.6 and will also mention the architecture in the experimental section to enhance the clarity of the paper.
>
> > **Q2**: re: generalizability of the hyperparameter balancing diphone and monophone losses
>
> The optimal value of the hyperparameter $\alpha$ which represents the tradeoff between diphone loss and monophone loss can vary across different datasets.

---

> > ### Comment · Reviewer_4p8d · 2024-12-02
> > **Thanks the authors for rebuttal**
> >
> > The authors addressed some of my concerns. However, due to my concern about the novelty of the paper, I'll keep my score.

---

### Author Response · Authors · 2024-11-28

We sincerely appreciate your insightful review and feedback, as well as the time you have dedicated to reviewing our paper. As the author-reviewer discussion period deadline is approaching, we kindly ask if our responses have addressed your comments and concerns. If you have any further suggestions or feedback, we would be very delighted to continue the discussion with you.

---

### Note · Authors · 2024-12-23

I have read and agree with the venue's withdrawal policy on behalf of myself and my co-authors.